# Endorsement of gender stereotypes in gender diverse and cisgender adolescents and their parents

**Benjamin deMayo** *, **Shira Kahn-Samuelson, Kristina R. Olson**

Department of Psychology, Princeton University, Peretsman Scully Hall, Princeton, New Jersey, United States of America

* bdemayo@princeton.edu

## Abstract

Previous work has documented adolescents' gender stereotype endorsement, or the extent to which one believes men or women should embody distinct traits. However, understanding of gender stereotype endorsement in gender diverse adolescents—those who identify as transgender, nonbinary, and/or gender nonconforming—is limited. Gender diverse adolescents' experiences with gender raise the question of whether they endorse gender stereotypes with the same frequency as cisgender adolescents. In this study, we investigated three primary research questions: (1) if gender diverse (N = 144) and cisgender (N = 174) adolescents (13–17 years) and their parents (N = 143 parents of gender diverse adolescents, N = 160 parents of cisgender adolescents) endorse gender stereotypes; (2) whether these groups differed from one another in their endorsement of gender stereotypes; and (3) whether parents' gender stereotyping was related to either their adolescents' stereotyping and/or their adolescents' predictions of their parents' stereotyping. We found (1) that participants showed low amounts of stereotyping; (2) there were no significant differences between gender stereotype endorsement in gender diverse and cisgender adolescents (or between their parents), though parents endorsed stereotypes slightly less than adolescents; and (3) there was a small positive association between adolescents' stereotyping and their parents' gender stereotyping. We discuss the limitations of our methods, and the possibility that rates of explicit stereotype endorsement may be changing over time.

## Introduction

Gender is one of the most salient social categories, starting early in childhood and continuing into adulthood [1]. As a result, a large body of psychological research has set out to understand people's acknowledgment and endorsement of gender stereotypes. Since adolescence is a pivotal time in social, emotional and sexual maturation [2,3] understanding how adolescents, generally defined as young people between puberty and adulthood, conceptualize and endorse gender stereotypes is an especially interesting question [4].

Almost all research on adolescents' gender stereotyping has studied cisgender people, or those whose gender identity matches the sex assigned to them at birth. Less is known about

**Data Availability Statement:** The data are publicly available on the Open Science Framework: https://osf.io/yxs3r/

**Funding:** This work was supported by National Institute of Child Health and Human Development, HD092347 and National Science Foundation, SMA-

1837857/SMA-2041463 to K.R.O. K.R.O. also receives funding from the MacArthur Foundation. The funders had no role in study design, data collection and analysis, decision to publish, or preparation of the manuscript.

**Competing interests:** The authors have declared that no competing interests exist.

gender stereotypes in gender diverse adolescents (including binary transgender, nonbinary and gender nonconforming individuals), despite the growing number of youth identifying with this group [5,6] a. Is the development of gender stereotyping different in this population of young people different than in the cisgender samples that have typically been studied? The question is of both theoretical and practical import. Understanding how gender diverse adolescents conceptualize gender stereotypes could further our understanding of how one's own experience with gender (non)conformity relates to stereotyping. Further, given the increasing visibility of gender diverse youth in the public sphere, it is critical that their experiences be represented in research documenting the trajectory of gender development across the lifespan.

In the present study, we set out to answer this question by assessing gender stereotyping using a previously validated measure [7] in a large sample of gender diverse adolescents. Additionally, we collected data from a large sample of cisgender adolescents as a comparison sample in order to assess whether gender diverse adolescents' gender stereotyping differs from that of cisgender adolescents. Finally, to better understand the relation between parents' beliefs and adolescents' beliefs, we asked whether parents' gender stereotyping is associated with either adolescents' own stereotyping or adolescents' perceptions of their parents' stereotyping.

## Gender stereotyping in adolescence

Prior research on gender stereotyping in adolescence has yielded mixed evidence on the extent of adolescents' gender stereotyping. Some researchers have found adolescence to be a time of life when gender roles intensify markedly (e.g., [8]), and find, as a consequence, that adolescents tend to rigidly endorse gender stereotypes, even more so than children in late elementary or early middle school [9]. Making a similar prediction, others posit that repeated reinforcement learning from the social environment regarding gender roles results in continued gender stereotype endorsement well into adolescence [10,11]. Conversely, some studies report an opposite trend of gender flexibility in adolescence [12,13], while others find that substantial individual differences obscure any clear group-level pattern of gender stereotyping in adolescence [14].

## Gender stereotyping in gender diverse youth

Regardless of how one construes prior research on adolescents' endorsement of gender stereotypes, the findings cannot confidently be applied to gender diverse individuals. In fact, until relatively recently, little empirical work had examined how transgender or other gender diverse youth of any age conceptualized gender stereotypes and whether they would endorse them in a meaningfully different way from cisgender peers. Three recent studies have, however, assessed gender stereotyping in transgender children, their siblings, and matched cisgender participants.

These studies—all of which studied younger children, not adolescents—show mixed evidence, but generally suggest cisgender and gender diverse children do not differ in their level of gender stereotyping. Three- to five-year-old transgender and cisgender children (siblings of transgender children and unrelated cisgender children) did not significantly differ in how much they thought men and women should engage in certain gender-stereotyped activities [15]. Similarly, 6–11-year-old transgender children, their siblings, and unrelated cisgender children did not differ significantly in their endorsement of prescriptive gender stereotypes; moreover, all groups of children tended to tolerate gender nonconformity [16].

However, a study focusing on gender stereotyping in 6–8-year-old children found that transgender children and their siblings showed significantly lower levels of gender stereotype endorsement, and more willingness to socially affiliate with gender nonconformers, than the

matched cisgender group [17]. In sum, preschool and elementary aged transgender children appear to endorse gender stereotypes at similar levels as their cisgender peers; when differences do appear, the transgender group appears to show lower levels of stereotype endorsement and greater tolerance of gender nonconformity. Our investigation of adolescents' gender stereotyping thus adds another data point that can help elaborate any possible between-group differences in stereotype endorsement (or lack thereof).

### Parent influence on adolescents' endorsement of gender stereotyping

In the current work, we were also interested in understanding whether adolescents' endorsement of gender stereotypes is associated with their parents' endorsement of gender stereotypes. Previous evidence has shown that parents' gender-related cognitions are associated with their children's. Notably, a meta-analysis [18] examined 43 studies investigating the link between parents' and children's gender schemas, and found that parents' gender-related attitudes about others were modestly associated with their children's ($r$ between 0.1 and 0.2). While the measures used, and psychological constructs assessed, in previous work vary considerably (see [19] for a review on measurement differences), past work generally indicates that parents may influence their children's thinking about gender [18–20]; we were thus interested in seeing whether we would obtain a similar result when examining gender diverse adolescents' stereotype endorsements.

### Current work

In our current work, we were interested in exploring the extent to which gender diverse adolescents endorse gender stereotypes, and how this may or may not differ from cisgender adolescents' gender stereotype endorsement. To do so, we recruited both a large sample of gender diverse adolescents as well as a group of cisgender adolescents, both of which completed a common measure of gender stereotyping (adapted from the OAT-AM; [7]) asking them to indicate how much they believed certain traits should be held by men versus women.

The OAT-AM (along with its equivalent for younger children, the COAT-AM) is a common measure of gender stereotyping in this age group. Studies using this measure have generally found that (presumably, primarily cisgender) adolescents show gender stereotyping (e.g., [21,22]. The OAT-AM carries several advantages which motivated its use in the current work. First, it is a short form that takes little time to complete and can be embedded in a larger study, as was the case here. Second, the scale is high in face validity, in that it probes participants directly on their endorsement of gender stereotypes. It also has good test-retest reliability [7].

Prior research motivates a variety of predictions with regards to group differences in gender stereotyping between gender diverse and cisgender youth. As discussed above, some prior studies have demonstrated that prepubescent gender diverse and cisgender children show similar levels of gender stereotyping, indicating that the same might be true of the adolescents studied in the current work [15,16]. Conversely, one might expect that gender diverse adolescents would show less rigidity in their attitudes about gender, as has sometimes been observed [17] (though note that in this work, siblings of transgender youth also showed less gender stereotyping, suggesting that other factors within a household might also contribute to reduced gender stereotyping).

We also asked adolescents' parents to complete the same gender stereotype endorsement measure as the adolescents and asked adolescents to complete the measure a second time, indicating their predictions of their parents' responses. The parent measure allowed us to investigate the exploratory question of whether parents of gender diverse adolescents would show different levels of gender stereotype endorsement than parents of cisgender adolescents, as well as to examine the relationship between parents' responses and their adolescents'. In

particular, we were interested in whether parents would show more or less overall stereotyping than their adolescents, and whether parents' stereotyping would be correlated with their adolescents' stereotyping.

Prior research that would inform predictions regarding group differences in gender stereotyping among parents is scarce. Some previous studies have speculated that parents of gender diverse children may engage their children in interactions that highlight flexibility in gender roles and communicate that it is acceptable to violate gender norms, as suggested by some findings that transgender children and their siblings tend to show more tolerance of gender nonconformity than cisgender children [17]. If such speculations are correct, one might expect parents of gender diverse youth to show less gender stereotyping than parents of cisgender youth, and that this apparent flexibility in the parents' views might be correlated with children and adolescents' own views on gender stereotypes, perhaps more so than in family units with cisgender children. However, previous research has not directly probed this question. Therefore, the extent to which parents of gender diverse youth might endorse stereotypes differently from parents of cisgender youth—and whether such a difference, if it exists, is related to their children's own gender stereotype endorsement—remains an open question.

Finally, we asked adolescents to predict their parents' responses on the measure. With this measure, we were interested in determining if parents' stereotype endorsement, adolescents' predictions about parents' stereotype endorsement, neither, or both were predictive of adolescents' responses on the same measure. We know of no past work speaking to this question and therefore included it as an additional exploratory research question.

## Methods

### Participants

Participants in this study were either part of the *gender diverse group* (after exclusions, *N* = 144 adolescents, *N* = 143 parents) or the *cisgender group* (after exclusions, *N* = 174 adolescents, *N* = 160 parents). Full parent demographic information can be found in Table 1; full adolescent demographic information can be found in Table 2.

**Determining whether adolescents were gender diverse or cisgender.** As we describe below, we recruited *gender diverse* and *cisgender* adolescents (and their parents) from different channels (which we refer to as the *gender diverse recruitment group* and the *cisgender recruitment* group respectively). In the vast majority of cases, adolescents from the *gender diverse recruitment group* were gender diverse, and adolescents from the *cisgender recruitment group* were cisgender. However, 4 adolescents from the *gender diverse recruitment group* identified as cisgender at the time of testing, and 8 adolescents from the *cisgender recruitment group* identified as transgender, gender nonconforming, or nonbinary, thus qualifying for our purposes as gender diverse at the time of testing. Henceforth, we use adolescents' own gender identification at the time of the study to determine whether they were counted as part of the *gender diverse group* or as part the *cisgender group*.

**Gender diverse group: Adolescents (N = 144).** Of the gender diverse adolescents included in this study, 72 are participants the research team had had prior contact with as gender diverse participants in larger longitudinal projects on gender development in U.S. and Canadian transgender or other gender diverse children. These youth were recruited through a variety of different sources including at camps and conferences for gender diverse youth, through medical and mental health providers, via word of mouth and in response to media stories, and through parents' online searches. These youth have been reported in several past papers about gender development [23–29] and about mental health [30–34]. The current measures were given as part of one wave of data collection. Of these gender diverse adolescents in

**Table 1. Parent demographics.**

| | Parents: Gender diverse group | Parents: Cisgender group | Difference Among Groups |
|---|---|---|---|
| Gender [a] | | | $\chi^2 = 15.877$[b], $p < .001$ |
| Woman | 118 (83%) | 155 (97%) | |
| Man | 19 (13%) | 4 (3%) | |
| Other gender or not reported | 6 (4%) | 1 (1%) | |
| Race | | | $\chi^2 = 0.313$[c], $p = 0.576$ |
| Asian | 3 (2%) | 10 (6%) | |
| Black/African | 1 (1%) | 1 (1%) | |
| Hispanic/Latino | 5 (3%) | 4 (3%) | |
| Multiracial/Other | 13 (9%) | 17 (11%) | |
| No race reported | 3 (2%) | 0 (0%) | |
| White/European | 118 (83%) | 128 (80%) | |
| Yearly income | | | $t(227.6)$[d] $= -5.21$, $p < .001$ |
| < \$25, 000 | 6 (4%) | 1 (1%) | |
| \$25,001-\$50,000 | 22 (15%) | 3 (2%) | |
| \$50,000-\$75,000 | 15 (10%) | 11 (7%) | |
| \$75,001-\$125,000 | 40 (28%) | 45 (28%) | |
| > \$125, 001 | 57 (40%) | 96 (60%) | |
| No income reported | 3 (2%) | 4 (3%) | |
| Mean politics rating (1 = most liberal, 7 = most conservative) | 1.91 | 2.59 | $t(293) = 4.519$, $p < .001$ |

Notes:

a. More detailed breakdown of participant gender in Supporting Information.

b. $\chi^2$ analysis compares distribution of gender between parents of gender diverse adolescents and parents of cisgender adolescents. For $\chi^2$ analysis on gender, participants were binned into categories of "women" and "other" due to small participant N's for men and individuals of other genders and the associated constraints for $\chi^2$ analyses.

c. $\chi^2$ analysis compares distribution of ethnicity between parents of gender diverse adolescents and parents of cisgender adolescents. For $\chi^2$ analysis on race, participants were binned into categories of "white" and "non-white" due to small participant N's in some ethnic/racial categories.

d. $t$-statistic derived from a 2 independent samples $t$-test in which each participant's income value was converted to a 1–5 scale (e.g., < \$25,000 ~ 1, \$25,001 - \$50,000 ~ 2, etc.). The negative value of the $t$-statistic is interpreted to indicate that parents in the *cisgender* group reported, on average, higher income levels than those in the *gender diverse* group.

the larger longitudinal projects, 4 had participated in a study on gender stereotyping that has been previously published [16]; no other previous publications examining stereotyping include participants in the current work.

On top of these participants with whom the researchers had already included as gender diverse participants in the larger study, there were 72 other adolescent participants in the *gender diverse* group. In order to expand the sample of gender diverse adolescents for this sample, 64 additional adolescents were recruited through email advertisements to listservs of professional organizations related to transgender health and well-being, parent listservs, and via social media and included as participants in the *gender diverse* group. Additionally, 3 participants who had previously participated as cisgender children in the aforementioned longitudinal projects identified as gender diverse at the time of the study, as did 5 other participants who had been specifically recruited with the intention of being in the *cisgender* group in this study (i.e., had not participated in prior studies from this research group). Altogether, the final sample size in the *gender diverse group* was 144 adolescents.

In addition to the participants above who were included in analyses, we received responses from 16 additional adolescent subject ID's in the *gender diverse* group which were excluded

**Table 2. Demographic breakdown of adolescent participants.**

| | Adolescents:<br>Gender diverse group | Adolescents:<br>Cisgender group | Difference Among<br>Groups |
|---|---|---|---|
| Race | | | $\chi^2 = 0.115^a$, $p = 0.734$ |
| Asian | 5 (3%) | 5 (3%) | |
| Black/African | 3 (2%) | 0 (0%) | |
| Hispanic/Latino | 4 (3%) | 3 (2%) | |
| Multiracial/Other | 26 (18%) | 42 (24%) | |
| White/European | 106 (74%) | 124 (71%) | |
| Gender [b] | | | $\chi^2 = 45.931^c$, $p < .001$ |
| Boy | 60 (42%) | 81 (47%) | |
| Girl | 49 (34%) | 92 (53%) | |
| Nonbinary or other | 35 (24%) | 1[d] (1%) | |
| Mean age (years) | 14.53 | 14.53 | $t(298) = 0.006$, $p = 0.995$ |

Notes:

a. $\chi^2$ analysis compares distribution of race between gender diverse adolescents and cisgender adolescents. For $\chi^2$ analysis on race, participants were binned into categories of "white" and "non-white" due to small participant N's in some ethnic/racial categories.

b. More detailed breakdown of participant gender in Supporting Information.

c. $\chi^2$ analysis compares distribution of gender between gender diverse adolescents and cisgender adolescents.

d. This participant gave a nonsense answer ("attack helicopter"), but other answers and the recruitment approach used for this participant led us to categorize them as a cisgender participant.

from analyses. During data collection, we developed data quality concerns emerging from a small number of the new gender diverse participants recruited from online channels–the only participants with whom the research team had not previously communicated; we therefore reviewed all of these non-longitudinal participants and decided on several exclusion criteria motivated by concerns about false participants (i.e., trolls or bots). All exclusions occurred without looking at the data of interest and were based on implausible inconsistencies in responding. Participants were excluded if (a) multiple consent/assent forms (e.g., the child and their parent) about the same adolescent listed different birth dates (excluded $N = 2$), (b) a participant reported that the adolescent was assigned male at birth but used only she/her pronouns at birth, or that the adolescent was assigned female at birth but used only he/him pronouns at birth (excluded $N = 8$), (c) the age of the adolescent did not match the reported birthdate (excluded $N = 1$), (d) parent and adolescent disagreed entirely on which pronouns were used to refer to the adolescent at equivalent times in their life-span (excluded $N = 5$; some variation in responses was tolerated, as in cases where a child recalled switching from "he" to "she" a year earlier than parents indicated, but complete deviations were not).

Beyond these adolescents excluded for quality control concerns, 6 additional adolescents recruited into the *gender diverse* group were excluded for not completing the OAT measures. In all, out of the 166 adolescent subject ID's we started with in the *gender diverse* group, we included data from 144.

**Gender diverse group: Parents (N = 143).** Parents of gender diverse youth were recruited into the study jointly with their adolescents. We began with survey responses from 198 subject ID's in the *gender diverse parents* group. Sixteen of these were the parent surveys associated with the 16 subject ID's in the *gender diverse adolescents* group which we excluded for quality control concerns (described above); additionally, another survey response from the *gender diverse parents* group (which did not have an adolescent response paired with it) was excluded for discrepant consent information. Thus, there were a total of 17 subject ID's in the *gender*

*diverse parents* group that were excluded for quality control concerns. On top of these quality control exclusions, 38 parents recruited into the *gender diverse* group were excluded because they did not have a child who completed a valid administration of the survey (these participants were excluded because this was primarily a study about adolescents' gender stereotyping); some of these 38 participants also met exclusion criteria for completing the survey too quickly (the full survey included other measures and had a median duration of 24 minutes; 2 parents in the *gender diverse* group were excluded for completing the survey in less than 5 minutes) or not completing the OAT measure (6 parents in the *gender diverse* group). In all, out of the 198 parents in the *gender diverse* group we began with, we had a final *N* of 143 parents (one parent in the *gender diverse* group had 2 adolescents participate, hence why the *N* for parents is one less than the *N* for adolescents in the *gender diverse* group).

**Cisgender group: Adolescents (N = 174).** Some of the cisgender adolescents in this research are part of the same longitudinal study as the transgender adolescents (N = 71); of these, 67 had previously participated as cisgender comparison participants in prior studies in the larger longitudinal project, and 4 had previously participated as gender diverse participants but identified as cisgender at the time of this study. The 65 adolescents who had previously participated as cisgender comparison participants in the larger longitudinal project were recruited in the past from the Communications Studies Participant Pool at the University of Washington. Of these cisgender adolescents, 4 had participated in a study on gender stereotyping that has been previously published (Rubin et al., 2020).

On top of the 71 adolescents who had previously been part of the larger longitudinal study (either as cisgender or gender diverse participants in the past), we recruited a sample of new cisgender adolescents (N = 103) from the Communications Studies Participant Pool to increase the sample size of the current study.

In addition to the above cisgender adolescents who we included in our analyses, 4 were excluded because they had not completed the OAT measure, and 1 was excluded because of a mismatch between their reported age and their birthdate. Thus, we started with 179 adolescent participants in the *cisgender* group, and had a final *N* of 174 adolescent participants in this group after exclusions.

**Cisgender group: Parents (N = 160).** Parents of cisgender youth were recruited into the study jointly with their adolescents. We began with 181 parent participants in the *cisgender* group. Of these, 20 were excluded because they did not have a child who completed a valid administration of the OAT measure (of these 20, 4 had not completed the OAT measure themselves), and 1 other was excluded from the analysis because of a mismatch between their child's birthdate and their reported age. Thus, we ended up with a sample size of 160 parents in the *cisgender* group after exclusions.

Of the parents in the *cisgender* group, 12 had two adolescents participate, and 1 had three adolescents participate. Thus, there were 160 parents in the *cisgender* group, 14 fewer than the number of adolescents in the *cisgender* group (N = 174).

**Siblings.** Some of the adolescent participants described above were siblings of other adolescents who also participated (N = 27 *cisgender* participants, N = 2 *gender diverse* participants). In these cases, parents often filled out the survey two or more times; if they did, we used the survey they completed first and associated it with both siblings, dropping the subsequent submissions.

## Procedure

Parents were sent the study materials via email. After giving consent for themselves and their children to participate, they completed the parent portion of the study. Adolescents could

either complete their portion immediately after the parent was finished on the same device, or they could opt to receive a follow-up email with the study materials. In either case, the parent completed their portion first so that they could consent to their own and their child's participation. The study procedure was approved by IRB protocol #00000157 at the University of Washington.

Participants completed these measures as part of a larger survey that investigated a range of different topics (e.g., mental health, medical transition, etc.). Survey completion took place between April 2019 and April 2020. The present measure was included as a stand-alone measure and therefore its relation to any other measures, beyond the demographics reported in this paper, has not been assessed.

## Measure

The trait subscale of the OAT-AM asks whether respondents think men, women, or both men and women should have various traits. In our adaptation, participants were shown 25 such traits. Ten traits were designated as stereotypically masculine (e.g., being good at math; being aggressive), ten as stereotypically feminine (e.g., crying a lot, being good at English), and five were gender neutral (e.g., study hard). (Masculine and feminine traits are listed in the *Results* section below, Table 3). Participants rate each trait on a 1 to 5 scale, with 1 indicating that only men should have the trait, 2 indicating that "mostly men, some women" should have the trait, 3 indicating that both women and men should have the trait, 4 indicating that "mostly women,

**Table 3. Items on the trait subscale of the OAT-AM with means, standard deviations, and the number of participants who skipped each item.**

| Item | Gender | Domain[a] | Adolescent self-report | | | Parent self-report | | | Adolescent prediction about the parent | | |
|---|---|---|---|---|---|---|---|---|---|---|---|
| | | | Mean | SE | Skipped (n) | Mean | SE | Skipped (n) | Mean | SE | Skipped (n) |
| be emotional | feminine | personality | 3.151 | 0.025 | 1 | 3.083 | 0.02 | 2 | 3.197 | 0.028 | 9 |
| be affectionate | feminine | personality | 3.08 | 0.021 | 4 | 3.056 | 0.016 | 0 | 3.104 | 0.024 | 9 |
| be good at English | feminine | academic | 3.039 | 0.018 | 8 | 3.034 | 0.014 | 11 | 3.052 | 0.022 | 8 |
| enjoy English | feminine | academic | 3.071 | 0.019 | 10 | 3.031 | 0.013 | 14 | 3.045 | 0.018 | 9 |
| be cruel | masculine | personality | 3.136 | 0.037 | 75 | 3.096 | 0.034 | 126 | 3.219 | 0.038 | 62 |
| be talkative | feminine | personality | 3.117 | 0.024 | 3 | 3.077 | 0.021 | 17 | 3.149 | 0.027 | 9 |
| be good at PE | masculine | academic | 3.096 | 0.026 | 6 | 3.021 | 0.015 | 13 | 3.127 | 0.027 | 12 |
| enjoy PE | masculine | academic | 3.091 | 0.027 | 11 | 3.017 | 0.014 | 12 | 3.114 | 0.026 | 12 |
| be gentle | feminine | personality | 3.189 | 0.029 | 1 | 3.096 | 0.021 | 1 | 3.196 | 0.029 | 7 |
| complain | feminine | personality | 3.073 | 0.029 | 31 | 3.016 | 0.02 | 54 | 3.07 | 0.03 | 34 |
| enjoy math | masculine | academic | 3.01 | 0.022 | 10 | 2.976 | 0.015 | 9 | 3.033 | 0.022 | 12 |
| be good at math | masculine | academic | 3.006 | 0.02 | 7 | 3.007 | 0.017 | 10 | 2.987 | 0.022 | 12 |
| be dominant | masculine | personality | 3.184 | 0.033 | 13 | 3.121 | 0.028 | 47 | 3.162 | 0.035 | 16 |
| cry a lot | feminine | personality | 3.28 | 0.032 | 14 | 3.302 | 0.035 | 78 | 3.272 | 0.034 | 20 |
| be neat | feminine | personality | 3.142 | 0.023 | 2 | 3.058 | 0.021 | 10 | 3.142 | 0.025 | 8 |
| act as a leader | masculine | personality | 3.035 | 0.024 | 1 | 3.02 | 0.019 | 3 | 3.064 | 0.023 | 6 |
| try to look good | feminine | personality | 3.13 | 0.027 | 11 | 3.06 | 0.022 | 21 | 3.182 | 0.029 | 10 |
| be good at science | masculine | academic | 3.016 | 0.021 | 6 | 3 | 0.015 | 11 | 2.993 | 0.023 | 11 |
| enjoy science | masculine | academic | 3.006 | 0.021 | 7 | 3.003 | 0.015 | 11 | 3.006 | 0.017 | 10 |
| be brave | masculine | personality | 3.06 | 0.026 | 1 | 2.983 | 0.019 | 3 | 3.042 | 0.026 | 6 |

Notes.

a. The distinction of academic vs. personality "domains" is not present in Liben & Bigler (2002) which first published and validated the trait subscale of the OAT-AM; however, we include it here since it corresponds to an exploratory analysis detailed in the Supporting Information (Section 6).

some men" should have the trait, and 5 indicating that only women should have the trait. In the original version of the measure, participants could indicate that "neither men nor women" should have particular traits; however, in this study we excluded that option because it seemed irrelevant to most responses, and we were concerned all negative traits might receive no codable responses as a result. However, we added an option to skip items if participants wanted to do so. For our analysis, masculine traits were reverse-coded, so that for all items, a score of 1 signified gender stereotype endorsement that is incongruent with societal expectations (i.e., men should cry a lot), while a score of 5 signified maximal gender stereotype endorsement congruent with societal expectations (i.e., men should be good at math). Gender neutral items were dropped from analyses for all participants. Skipped items were excluded from the computation of individual participants' mean scores.

We obtained 3 final scores on the trait subscale of the OAT-AM per parent-adolescent dyad, each ranging from 1 (strong counter-stereotypical responding) to 5 (strong stereotypical responding): the *adolescent self-report* measure, the *parent self-report* measure, and the *adolescent prediction about the parent* measure. In the latter, the adolescent was asked to indicate how they thought their parent would respond to the trait subscale of the OAT-AM. Cronbach's $\alpha$ for the *parent self-report*, *adolescent prediction about the parent*, and *adolescent prediction about the parent* measures were 0.64, 0.78, 0.81 respectively; the low inter-item reliability on the *parent self-report* measure is discussed further in the discussion section.

## Primary research questions

We investigated three primary research questions: (1) whether participants showed gender stereotyping; (2) whether there were group differences (both cisgender vs. gender diverse, as well as adolescents vs. parents) in gender stereotype endorsement; and (3) whether adolescents' gender stereotype endorsement, and their predictions about their parents' gender stereotype endorsement, were related to parents' gender stereotype endorsement.

## Results

### Gender stereotype endorsement

First, we investigated whether adolescents and their parents showed evidence of gender stereotyping. For each participant, we calculated their average self-reported stereotype endorsement score by taking the mean of their responses on the *adolescent self-report* measure for adolescents and the *parent self-report* measure for the parents. A one-sample *t*-test revealed that, averaging across adolescents and parents, participants' mean gender stereotyping scores ($\mu$ = 3.071, SD = 0.162) were significantly greater than the null value of 3, $t(620) = 10.942$, $p < .001$, Cohen's $d$ = 0.439, indicating that participants endorsed gender stereotypes in a direction that was congruent with societal expectations. However, examination of the actual mean (3.07 on a 5-point scale) indicates that this was a very small tendency overall. Further, as shown in Fig 1, all groups indicated that "both men and women" should have the stereotypically masculine and feminine traits more than 80% of the time, suggesting that explicit endorsement of stereotypes on the trait subscale of the OAT-AM was relatively rare.

While overall levels of stereotyping were low, individual items on the OAT-AM varied both in their mean endorsement and in how much response variability they displayed across participants (Table 3). Tables 4 and 5 show means and standard errors of stereotyping scores for items on the trait subscale of the OAT-AM, broken down by the gender of the stereotype (feminine vs. masculine) and domain of the stereotype (academic/extracurricular vs. personality) respectively. Exploratory statistical analyses probing both of these effects are in the Supporting information (Sections 5 and 6).

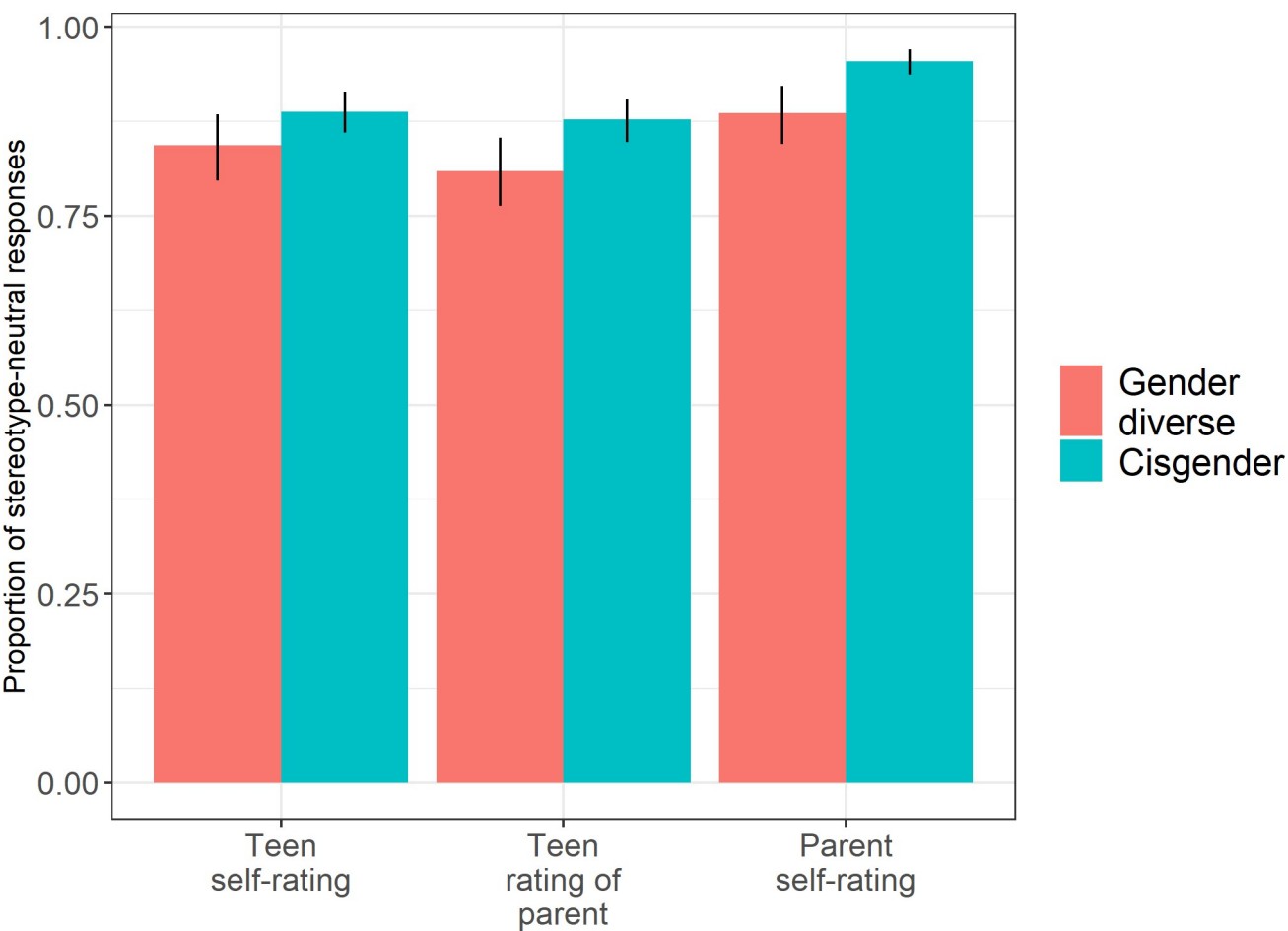

**Fig 1. Proportion of non-stereotyped responses (i.e., "both women and men"), by gender identity condition (either *gender diverse*, *N* = 144 dyads, or *cisgender*, N = 174 dyads).** Some parents appeared in multiple dyads.

## Group differences in gender stereotype endorsement

Next, we were interested in whether gender diverse adolescents and their parents endorsed gender stereotypes at differing levels from cisgender adolescents and their parents. We fit a linear mixed-effects model predicting participants' average gender stereotype endorsement scores as a function of the gender identity of the adolescent in the dyad (*gender diverse* or *cisgender*), whether the respondent was an adolescent or a parent, and the interaction between these two factors. In order to account for the fact that each family has multiple mean scores (one for the *parent self-report* and at least one other for the *adolescent self-report*, and

**Table 4. Means and standard errors of stereotype endorsement on feminine and masculine items of the trait subscale of the OAT-AM.**

| Measure | Mean (SE) | |
|---|---|---|
| | **Feminine items** | **Masculine items** |
| Adolescent self-rating | 3.13 (0.014) | 3.06 (0.016) |
| Parent self-rating | 3.07 (0.011) | 3.02 (0.011) |
| Adolescent predictions about the parent | 3.14 (0.016) | 3.07 (0.016) |

**Table 5. Means and standard errors of stereotype endorsement on academic/extracurricular-related items and personality-related items of the trait subscale of the OAT-AM.**

| Measure | Mean (SE) | |
|---|---|---|
| | **Academic/extra-curricular items** | **Personality items** |
| Adolescent self-rating | 3.04 (0.011) | 3.13 (0.014) |
| Parent self-rating | 3.01 (0.008) | 3.07 (0.010) |
| Adolescent predictions about the parent | 3.05 (0.011) | 3.15 (0.015) |

occasionally more if a family had multiple adolescents participate), we included "random" intercepts for each family. (In Section 1 of the Supporting Information, we show two alternative methods of analyzing the results in which we treat participants' responses as a binary variable; we include these analyses to adhere more closely to the recommended scoring procedure recommended by [7]. The results are similar across either analytic approach.)

Tables 6 and 7 summarize the overall results. We found no significant differences in gender stereotyping on the basis of gender identity; stereotyping in participants from the *gender diverse group* did not differ from stereotype endorsement from those in the *cisgender group*, $\beta$ = -0.01, $p$ = 0.61. However, the mixed-effects regression model does show a significant main effect such that parents' responses are slightly lower in stereotype endorsement than adolescents', $\beta$ = -0.056, $p < 0.001$, corresponding to a reduction of 0.34 standard deviations in stereotype endorsement.

As an exploratory sub-analysis, we also examined whether any differences in gender stereotyping emerged between parents (Table 8) and adolescents (Table 9) of different genders. An independent-sample $t$-test comparing gender stereotype endorsement scores of parents who were men and parents who were women revealed that men had higher average gender stereotype endorsement scores than women, $t(294)$ = -2.642, $p$ = 0.009, echoing prior work that has suggested that fathers may hold more explicit gender stereotypes than mothers [35]. We also performed a one-way ANOVA to examine whether there were any differences in gender stereotype endorsement scores between boys, girls and a group consisting of adolescents who identified as nonbinary or another gender. Adolescents of different genders did not significantly differ from one another ($F$ (2, 315 = 2.889, $p$ = 0.057); however, exploratory post-hoc Tukey HSD comparisons showed that, while nonbinary or other adolescents did not differ from girls ($p$ = 0.72) or boys ($p$ = 0.73), girls may have been slightly less likely to endorse stereotypes than boys ($p$ = 0.04).

## Relationship between parents' and adolescents' endorsement of gender stereotypes

Finally, we examined whether parents' stereotype endorsement (*parent self-report* measure) was associated with adolescents' stereotype endorsement (*adolescent self-report* measure) or adolescents' predictions of their parents' stereotyping (*adolescent prediction about the parent*

**Table 6. Results from linear mixed-effect model predicting participants' average gender stereotype endorsement scores.** Reference group is cisgender adolescent self-report.

| Predictor | Estimate | Standard Error | df | *t*-value | *p*-value |
|---|---|---|---|---|---|
| Intercept | 3.10 | 0.01 | 581.62 | 252.85 | < .001 |
| Parent self-report (vs. adolescent) | -0.06 | 0.02 | 319.03 | -3.35 | < .001 |
| Gender diverse group (vs. cisgender) | -0.01 | 0.02 | 596.74 | -0.52 | 0.605 |
| Parent self-report * Gender diverse group | 0.02 | 0.02 | 314.96 | 0.85 | 0.395 |

**Table 7. Means, standard errors, and N's by participant group and measure.**

| Measure | Gender Diverse Group | | Cisgender group | |
|---|---|---|---|---|
| | Mean (SE) | N | Mean (SE) | N |
| Adolescent predictions about the parent | 3.11 (0.018) | 141 | 3.11 (0.016) | 171 |
| Parent self-rating | 3.05 (0.011) | 143 | 3.04 (0.007) | 160 |
| Adolescent self-rating | 3.09 (0.018) | 144 | 3.10 (0.013) | 174 |

**Table 8. Parent stereotype endorsement: Means, standard errors, and N's by parent gender.**

| Parent Gender | Mean (SE) | N |
|---|---|---|
| Woman | 3.04 (0.006) | 273 |
| Man | 3.11 (0.046) | 23 |
| Nonbinary, other or not reported | 3 (0) | 7 |

**Table 9. Adolescent stereotype endorsement: Means, standard errors, and N's by adolescent gender.**

| Adolescent Gender | Mean (SE) | N |
|---|---|---|
| Girl | 3.07 (0.013) | 141 |
| Boy | 3.12 (0.017) | 141 |
| Nonbinary or other | 3.09 (0.047) | 36 |

**Table 10. Results from linear regression predicting adolescents' mean stereotyping scores as a function of their parents' scores.**

| Predictor | Estimate | Std. Error | t-value | p-value |
|---|---|---|---|---|
| Intercept | 2.440 | 0.29 | 8.29 | < .001 |
| Parent self-report | 0.23 | 0.10 | 2.39 | 0.02 |

**Table 11. Results from linear regression predicting adolescents' mean predictions about their parents as a function of their parents' scores.**

| Predictor | Estimate | Std. Error | t-value | p-value |
|---|---|---|---|---|
| Intercept | 2.48 | 0.33 | 7.55 | < .001 |
| Parent self-report | 0.21 | 0.11 | 1.93 | 0.05 |

measure). To examine whether adolescents' stereotyping was associated with their parents', we ran a linear regression predicting adolescents' scores from their parents' scores. This analysis revealed a very small but significant effect of parent stereotyping score, $\beta = 0.23$, $t = 2.385$, $p = 0.018$ (Table 10). Similarly, to examine whether the adolescents' predictions of their parents stereotyping was predictive of their parents' actual stereotyping, we ran a linear regression predicting adolescents' predictions about their parents as a function of the parent's actual stereotyping (Table 11). This analysis did not reveal a significant effect of parent stereotyping score, $\beta = 0.21$, $t = 1.93$, $p = 0.05$. Both of the aforementioned analyses include some redundancy in the data because some parents had multiple children participate. To account for this nonindependence, we attempted to fit linear mixed-effects models with random intercepts for

each family, but since these mixed-effects models obtained singular fits, we instead report the simpler linear models. Estimates of regression coefficients were almost identical between these mixed-effects models and the linear models we report here.

In addition to these analyses, we also conducted exploratory regression analyses to examine whether the relationship between *parent self-report* and either of the measures completed by the adolescents might be stronger in the *gender diverse* or *cisgender* group. Given the overall lack of difference between groups and the exploratory nature of these analyses, we refer readers to the Supporting Information (Section 4) for those results. We also include a correlation table illustrating Pearson's *r* correlations between all three of the outcome stereotyping measures (*adolescent self-report*, *parent self-report*, and *adolescent prediction about the caregiver*) in the Supporting Information (Section 7).

## Discussion

We used a previously validated measure [7] that has historically resulted in significant levels of gender stereotyping (e.g, [21]) to assess gender stereotype endorsement in gender diverse and cisgender adolescents, as well as their parents. Two main findings emerged. First, even though the mean gender stereotype endorsement score across participants was significantly higher than the null value (which would have indicated a complete lack of stereotype endorsement), all groups of adolescents and parents showed remarkably little endorsement of gender stereotypes (Fig 1). On every item of the trait subscale of the OAT-AM, at least 67% of participants who responded to the item endorsed gender stereotype flexibility, indicating that 'both men and women' should show a particular trait (e.g., saying both men and women should be good at math); the median rate of choosing 'both men and women' across all items was 88%. Parents endorsed stereotypes even less on average than adolescents. Among adolescents, we observed no significant differences between gender diverse participants and cisgender participants. This finding converges with those in [15] and [16], in which gender diverse and cisgender children did not show differences in gender stereotyping, though differs from a prior study [17] which found that 6–8-year-old transgender children showed less gender stereotyping compared to unrelated cisgender children.

Second, we observed a small relationship between adolescents' stereotype endorsement and their parents' stereotype endorsement. The size and direction of this effect (small, but positive) is reflective of the more general phenomenon described in [18] that parents' thinking about gender is modestly correlated with their children's.

Apart from these main findings, exploratory analyses suggested that adolescent boys and parents who identified as men showed more gender stereotype endorsement than parents who identified as women and adolescent girls respectively. These results do not bear directly on our original research questions, but the finding regarding parental gender differences shows concordance with prior research [35].

Several limitations are present in the current work. First, the trait subscale of the OAT-AM, while face valid and used widely the study of gender stereotyping, including in adolescents (e.g., [22]), showed remarkably little response variability across participants, contributing to a low value of Cronbach's $\alpha$ on the *parent self-report* measure; in fact, when indicating their own stereotype endorsement, over half of participants (60% of adolescents, 66% of parents) responded to every item on the scale (excluding skipped items) by saying that "both men and women" should embody the trait in question, meaning that all of the effects observed were driven by fewer than half the participants in the sample. The scale may have been too coarse to show more nuanced group-level differences in endorsement of gender stereotypes, or these may not have been the best example traits for assessing gender stereotyping at this particular

moment in history. In future work, it may be more appropriate to use a measure that is less direct than the OAT-AM, since participants may be hesitant to explicitly deem certain traits as "man-like" or "woman-like." One possible way of avoiding this directness would be to assess participants' descriptive, rather than prescriptive, stereotypes (in other words, asking people what individuals of different genders *do* do, not what they *should* do). Another potentially interesting avenue for future research involves using implicit measures to probe gender-stereotyped attitudes in adolescents in their parents, since participants in the current social climate may not explicitly endorse (or even acknowledge) stereotypes to the same extent as participants in studies several decades ago.

Additionally, our participant sample is also skewed towards white, upper middle- and upper-class people in the United States of America who are politically liberal. As a result, the generalizability of these findings to a more representative sample of the U.S. population, or populations in other cultural or national contexts, is unknown. Participants in the *gender diverse* and *cisgender* groups were also not perfectly matched on demographics (Tables 1 and 2); in particular, parents in the *gender diverse* group were less affluent and more liberal than parents in the *cisgender* group. Given that we did not see stark differences in gender stereotype endorsement between groups, we believe it is unlikely that demographic discrepancies compromise the comparability of these samples. However, participants in the *cisgender* group were primarily drawn from a metropolitan area in the U.S. Pacific Northwest, a region that has a more progressive orientation on social issues than the country as a whole [36]. As such, it is possible that a more nationally representative sample of cisgender adolescents would have demonstrated a greater propensity to endorse gender stereotypes than the participants in this study.

Despite these limitations, we believe these findings also present a possible summary of how adolescents are in thinking about gender stereotypes today. Perhaps they are endorsing gender stereotypes less than past generations [22], an intriguing idea for follow-up research.

## Conclusion

We found that gender diverse adolescents and cisgender adolescents showed similar levels of endorsement of gender stereotype endorsement, suggesting that the experience of being gender diverse may not exert a strong influence on adolescents' propensity to endorse gender stereotypes. Adolescents' parents tended to show less gender stereotype endorsement than adolescents, but all groups' stereotype endorsement was low. To the extent that adolescents did endorse gender stereotypes, their stereotype endorsement showed a very slight positive association with their parents' stereotype endorsement. These results contribute to a growing body of empirical work that aims to understand how an increasingly visible cohort of transgender, gender nonconforming and nonbinary youth engage with prevailing societal stereotypes about gender.

## Supporting information

**S1 File.**
(DOCX)

## Acknowledgments

We thank Robin Sifre, Natalie Gallagher, and Dominic Gibson for their statistical guidance in preparing this manuscript.

## Author Contributions

**Conceptualization:** Kristina R. Olson.

**Data curation:** Benjamin deMayo, Shira Kahn-Samuelson.

**Formal analysis:** Benjamin deMayo, Kristina R. Olson.

**Funding acquisition:** Kristina R. Olson.

**Investigation:** Kristina R. Olson.

**Methodology:** Benjamin deMayo, Kristina R. Olson.

**Project administration:** Shira Kahn-Samuelson, Kristina R. Olson.

**Resources:** Kristina R. Olson.

**Supervision:** Kristina R. Olson.

**Validation:** Kristina R. Olson.

**Visualization:** Benjamin deMayo.

**Writing – original draft:** Benjamin deMayo.

**Writing – review & editing:** Benjamin deMayo, Shira Kahn-Samuelson, Kristina R. Olson.

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
