## [Decision Letter · Decision Letter 0]

28 Oct 2021

PONE-D-21-20601

Endorsement of gender stereotypes in gender diverse and cisgender adolescents and their parents

PLOS ONE

Dear Dr. deMayo,

Thank you for submitting your manuscript to PLOS ONE. I was fortunate enough to receive reviews from two experts in the field, and thoroughly read and assessed this paper myself. The reviewers provided very different assessments about the merits of this paper, with Reviewer 1 believing that this could be accepted pending minor revisions and Reviewer 2 expressing hesitation around whether this paper provides sufficiently original research (PLOS ONE publication criterion 1) and a sufficiently appropriate approach to warrant publication in this journal. I agree with the points raised by both of the reviewers. After careful consideration, I believe that this paper has merit and the potential to make a novel contribution to the field, but it does not yet meet PLOS ONE’s publication criteria as it currently stands. I believe that the focus on gender stereotyping among gender diverse and cisgender adolescents and their parents is unique and important; this paper will contribute to a growing literature outlining the experience and beliefs of gender diverse children and their families. Therefore, I would like to invite you to submit a revised version of the manuscript that addresses the points raised during the review process. That said, I must emphasize that I cannot guarantee that a revised manuscript will be accepted in this journal. In addition, I may decide to send this back out for another round of reviews.

I will not reiterate the points raised by the reviewers but will instead highlight a few of my own that should also be addressed in the revision. These include:

Please be sure to submit the revised manuscript in the appropriate format (https://journals.plos.org/plosone/s/submission-guidelines). As one example “Manuscript text should be double-spaced. Do not format text in multiple columns.” You might want to refer to the APA manual (7^th^ edition) for suggestions as well.On page two, you indicate that several recent studies have examined gender stereotyping in transgender youth, but then proceed to review papers with children. Please change the wording to reflect this. The more important question, raised also by Reviewer 1, is whether each of these studies were conducted with the same children. That is, are conclusions drawn about the gender stereotyping of gender diverse preschoolers, young children, and older children all using the same (or many of the same) participants as this study that is examining adolescents? If so, this should be clearly explained in the paper. Related to this, please clarify how many gender diverse adolescents were recruited from other sources.Given that the OAT-AM is almost 20 year old, it seems possible that some of the items are dated. It would be useful to know more about this measure and the results. Was most of the variability driven by some items (if so, which ones) and did some items not vary at all? It would be useful if the items could be detailed on OSF either in the data file (e.g., Parent_OAT_1_Math) and/or in the variable explanations. In addition, it would be ideal to provide not only the final data file, but also a data file that includes all participants (prior to exclusions) on OSF.In the final sentence of the abstract, you indicate that “these results suggest…” but none of the results reported to that point suggest what follows. This relates to your research questions and hypotheses – it does seem that one of the questions that you address is whether there will be explicit stereotype endorsement and not just a difference between the groups, and you might consider adding this as a research question when revising, as well as revising the abstract to address this inconsistency.The analyses in the supplement did not make sense to me, as participants’ responses were recoded based on whether they selected 3 (no gender difference) or not. However, the direction of that ‘not’ is important, as it reflects gender stereotyping, or counterstereotypical beliefs. I would recommend re-running these analyses, pitting stereotype-consistent responses against non-stereotype-consistent responses (e.g., 3 or counterstereotypical). Given the restricted variability this might not make any meaningful difference in the results, but this seems a more appropriate analytic approach given your research questions.My final point is that it would be ideal to expand the scope of this paper if possible. I wonder if there are additional questions that might be addressed, for example if some of these adolescents did provide responses at an earlier time that could be compared. Reviewer 2 provides several other suggestions using the current data (see points 6 & 7) or with additional data (see point 12). Given that the main finding is a lack of difference between groups combined with low variability and a general lack of stereotyping on an older measure, I believe that the findings would be more compelling with additional analyses, or ideally additional data. I hope that you will decide to revise and resubmit this paper and will look forward to receiving your re-submission. Please submit your revised manuscript by Dec 12 2021 11:59PM. If you will need more time than this to complete your revisions, please reply to this message or contact the journal office at plosone@plos.org. Please include the following items when submitting your revised manuscript:

We look forward to receiving your revised manuscript.

Kind regards,

Jennifer Steele

Academic Editor

PLOS ONE

Journal Requirements:

2. Your abstract cannot contain citations. Please only include citations in the body text of the manuscript, and ensure that they remain in ascending numerical order on first mention.

Reviewers' comments:

Reviewer's Responses to Questions

**Comments to the Author**

1. Is the manuscript technically sound, and do the data support the conclusions?

Reviewer #1: Yes

Reviewer #2: No

2. Has the statistical analysis been performed appropriately and rigorously? 

Reviewer #1: Yes

Reviewer #2: No

3. Have the authors made all data underlying the findings in their manuscript fully available?

Reviewer #1: Yes

Reviewer #2: Yes

4. Is the manuscript presented in an intelligible fashion and written in standard English?

Reviewer #1: Yes

Reviewer #2: Yes

5. Review Comments to the Author

Reviewer #1: One concern about dual publication is the use of the same gender diverse adolescents across multiple manuscripts. I know this is a large data set and it is reasonable that it will result in multiple manuscripts, but given how little research has been conducted with gender diverse youth, the repeated publication from one data set may skew the knowledge base. I recommend citing all of the other publications using the same youth in the Participants section.

Otherwise, the analyses are appropriate and the text is clear and complete. The conclusions are appropriate based on the results.

Reviewer #2: This paper investigated explicit gender stereotype endorsement in trans and cis adolescents and their parents. The sample is large and unique, and the topic is timely. The study employed an established measure of gender stereotyping. The study has potential to make a solid contribution to the literature, but I noted several aspects of the study and manuscript that should be addressed prior to further consideration for publication. Please see my comments below.

1. The Introduction could benefit from providing more of a theoretical framework, or at least providing more context to support hypotheses/predictions about why trans, compared with cis, adolescents (and their respective parents) would show more, less, or similar patterns of gender stereotyping. The authors should also give more background on parental gender socialization and associations between parent and adolescent gender stereotyping. Would the authors predict a difference in the strength of the association between trans vs. cis parent-adolescent dyads? If memory serves, Olson and Enright’s 2018 Dev Sci article reported that both trans children and their siblings were similarly accepting of gender nonconformity (compared with other cisgender controls). Could this reflect some differential parental socialization taking place with regard to acceptability of deviations from gender norms? There is also no explanation regarding the value of considering both parents’ actual gender stereotyping vs. adolescents’ perceptions of the extent to which their parents would gender stereotype. In short, the Introduction should be developed further to convey the theoretical and/or practical importance of the kinds of questions being asked here. To the extent that directional predictions (even if competing) can be offered, I encourage that as well.

2. I found that there were several statements of fact that were not backed up by a citation. Examples include claims that the numbers of trans adolescents are increasing and that past work generally finds that parents influence their children’s thinking about gender (regarding this latter one, I wonder whether the authors meant children and adolescents because the latter seems more relevant to concentrate on here). I suggest the authors make sure to back up statements of fact with appropriate citations.

3. What is the benefit of retaining participants who skipped all questions? Seems that if they did not provide data on the dependent variables, they should be dropped because they are leading to a skewed sense of the characteristics of the groups of participants who contributed the key data.

4. The samples do not seem to be matched on demographics. The cisgender sample appears to be more affluent and less white/more multiracial (based on the parent data). Are the authors concerned that this might be an important confound in their analyses? There is some research suggesting racial differences in gender development exist (e.g., work by May Ling Halim). Also, the cisgender controls were recruited from a participant database at the authors’ institution. In the present case, I assume this means that participants were from a relatively urban center in the US Pacific Northwest, which probably has a particular social climate. I wonder whether the gender diverse sample is from a more varied set of backgrounds given they were recruited from across the US and Canada, and whether this is also a relevant confound to consider in weighing the comparability of the participant groups.

5. Can the authors please say more about the recruitment method for online participants as well as for the other samples? I know this is part of a larger project and the authors have maybe shared some of these details elsewhere (especially on the longitudinal sample), but it’s not clear for those who are maybe only reading this paper from this team. Also, I wonder how this sample relates to prior ones that this group reported on with respect to gender stereotyping. There were three relevant studies from this team that were reviewed in the Introduction. Were the participants in this study the same as any of those prior ones? Or is this a completely different cohort?

6. The authors do not note the parent gender. There is literature suggesting mothers and fathers hold different attitudes about gender roles/stereotypes. I suggest reviewing that literature and analyzing by parent gender. Perhaps see work by Joyce Endendijk and colleagues on this topic.

7. The authors do not note the adolescent participants’ gender breakdown. Are there adolescent gender differences among cis samples in gender stereotyping? Any reason to suspect there might be differences between trans boys vs. trans girls vs. nonbinary, and so on? Even a preliminary analysis of this question would be important/interesting. In any case, it is presently unclear how comparable the samples are with respect to the gender composition with regard to cis/trans feminine/masculine individuals.

8. Can the authors please provide reliability analysis data on the OAT-AM for the present sample? Are all the items on this scale contributing to reliability? For example, there is some recent work suggesting that the stereotype that boys are superior at math is not always endorsed in adolescent samples (Morrissey et al 2019 in J Adolescence).

9. The authors state that the OAT-AM used in the current study was adapted lightly. Please explain.

10. It would be helpful to explain how the OAT-AM was scored in the Method section. A statistical analysis subsection would also be helpful to evaluate/understand the analytic approach. As is, the research questions and analyses are all somewhat vague, which makes it difficult to discern whether the optimal approach is being employed.

11. Tables 3: Please provide a more descriptive title.

12. A main finding is that participants, regardless of group or age, were unlikely to endorse prescriptive gender stereotypes. One wonders what might have happened had the authors measured descriptive stereotypes. I also wonder whether we are now in an era where people hold (or at least report) explicit views that run contrary to traditional gender stereotypes. Perhaps an implicit measure would yield some different results. Given this team’s expertise in this area, I would be interested to see some discussion of these possibilities folded into this paper.

13. Page 8, line 255: The authors are making the point that one study found trans children ages 6-8 years gender stereotyped less. But in the Introduction they noted that trans children that study were similar to their cis siblings. So, it seems a little dubious to me to claim in the Discussion that the study of 6-8 year-olds is finding something that suggests a trans vs. cis difference.

14. The figure quality appeared “fuzzy” on my end. Consider revising to higher resolution.

6. PLOS authors have the option to publish the peer review history of their article (what does this mean?). If published, this will include your full peer review and any attached files.

Reviewer #1: No

Reviewer #2: No

---

## [Author Response · Author response to Decision Letter 0]

28 Jan 2022

Please be sure to submit the revised manuscript in the appropriate format (https://journals.plos.org/plosone/s/submission-guidelines). As one example “Manuscript text should be double-spaced. Do not format text in multiple columns.” You might want to refer to the APA manual (7th edition) for suggestions as well.

Thank you, we have made sure the paper is appropriately formatted.

On page two, you indicate that several recent studies have examined gender stereotyping in transgender youth, but then proceed to review papers with children. Please change the wording to reflect this. The more important question, raised also by Reviewer 1, is whether each of these studies were conducted with the same children. That is, are conclusions drawn about the gender stereotyping of gender diverse preschoolers, young children, and older children all using the same (or many of the same) participants as this study that is examining adolescents? If so, this should be clearly explained in the paper. Related to this, please clarify how many gender diverse adolescents were recruited from other sources. 

Thank you for this suggestion. We have made the language more consistent to reflect the fact that prior research has been conducted with children under the age of 11, but that our study involves adolescents between the ages of 13 and 17. We have also clarified how many participants in the present work are included in previous studies (n=8), and have specified how many gender diverse adolescents were recruited from other sources (n=64).

Given that the OAT-AM is almost 20 year old, it seems possible that some of the items are dated. It would be useful to know more about this measure and the results. Was most of the variability driven by some items (if so, which ones) and did some items not vary at all? It would be useful if the items could be detailed on OSF either in the data file (e.g., Parent_OAT_1_Math) and/or in the variable explanations. In addition, it would be ideal to provide not only the final data file, but also a data file that includes all participants (prior to exclusions) on OSF.

Thank you for this suggestion. We have included a table in the Supplementary Material that shows the mean score for each item on the scale and its standard error. We have also included reliability analysis for the items on the trait subscale of the OAT-AM in response to a comment from R2. Finally, we have included explanations of each of the items on the trait subscale of the OAT-AM on OSF (wording of items as presented to participants can be found in `oat_items.csv`), as well as versions of the data file with participants prior to exclusions (titled `kid_data_full_osf.csv` for the parents and `parent_data_full_osf.csv`). All of these files can be found in the `resubmission` folder in the OSF page: https://osf.io/yxs3r/files/

In the final sentence of the abstract, you indicate that “these results suggest…” but none of the results reported to that point suggest what follows. This relates to your research questions and hypotheses – it does seem that one of the questions that you address is whether there will be explicit stereotype endorsement and not just a difference between the groups, and you might consider adding this as a research question when revising, as well as revising the abstract to address this inconsistency.

Thank you for this suggestion. We have added another research question and a corresponding subsection in the “Results” section that more directly probes whether there was explicit stereotype endorsement at all, not just a difference between the groups. We have also accordingly edited the abstract to reflect this change.

The analyses in the supplement did not make sense to me, as participants’ responses were recoded based on whether they selected 3 (no gender difference) or not. However, the direction of that ‘not’ is important, as it reflects gender stereotyping, or counterstereotypical beliefs. I would recommend re-running these analyses, pitting stereotype-consistent responses against non-stereotype-consistent responses (e.g., 3 or counterstereotypical). Given the restricted variability this might not make any meaningful difference in the results, but this seems a more appropriate analytic approach given your research questions. 

Thank you for this recommendation. We have re-run this analysis in the supplementary material with the gender stereotyping variable recoded as you suggest. We agree with you that knowing the direction matters. In addition, we retained the initial coding (based on whether they selected 3 or not) in the supplement as well, since this is the method of response scoring suggested by the original creators of the OAT-AM, and we want interested readers to be able to compare our results to that work.

My final point is that it would be ideal to expand the scope of this paper if possible. I wonder if there are additional questions that might be addressed, for example if some of these adolescents did provide responses at an earlier time that could be compared. Reviewer 2 provides several other suggestions using the current data (see points 6 & 7) or with additional data (see point 12). Given that the main finding is a lack of difference between groups combined with low variability and a general lack of stereotyping on an older measure, I believe that the findings would be more compelling with additional analyses, or ideally additional data.

Thank you for this suggestion. We agree that additional data would make the findings more compelling, and we looked into the possibility of correlating past data from the same participants with the stereotyping scores reported here; however, only 8 of the adolescents in this study had completed a stereotyping measure in the past . Thus, there was not enough longitudinal data on the present construct to make an interesting addition to the paper. 

Further, there are two other issues with adding more data from past administrations with this cohort. First, there is almost no variability in the present results meaning that it would be nearly impossible to observe any meaningful association between these results and anything else. If we move to an even further construct (e.g., peer preferences, self-esteem), we’d be even less likely to find an association. Second, many of the participants in the present work had never participated in a study with us before and therefore we would need to exclude them from such analyses, lowering our sample size, and further reducing the odds of finding significant associations over time.

We did, however, add some additional analyses of the present data in response to points 6 & 7 made by Reviewer 2 (examining gender stereotype endorsement by parents’ gender and adolescents’ gender); we also include an exploratory analysis in the Supplementary Materials that examines whether gender diverse adolescents’ stereotyping was more (or less) predictive of their parents’ stereotyping than cisgender adolescents’ stereotyping. 

Reviewer 1:

Reviewer #1: One concern about dual publication is the use of the same gender diverse adolescents across multiple manuscripts. I know this is a large data set and it is reasonable that it will result in multiple manuscripts, but given how little research has been conducted with gender diverse youth, the repeated publication from one data set may skew the knowledge base. I recommend citing all of the other publications using the same youth in the Participants section.

Otherwise, the analyses are appropriate and the text is clear and complete. The conclusions are appropriate based on the results.

Thank you for this suggestion. We have cited all the other publications using the same youth in the participants section. Importantly, only 8 of the transgender participants completed a past stereotyping measure. Sixty-four of the gender diverse participants and 107 of the cisgender participants have never completed any studies with our team.

Reviewer 2:

1. The Introduction could benefit from providing more of a theoretical framework, or at least providing more context to support hypotheses/predictions about why trans, compared with cis, adolescents (and their respective parents) would show more, less, or similar patterns of gender stereotyping. The authors should also give more background on parental gender socialization and associations between parent and adolescent gender stereotyping. Would the authors predict a difference in the strength of the association between trans vs. cis parent-adolescent dyads? If memory serves, Olson and Enright’s 2018 Dev Sci article reported that both trans children and their siblings were similarly accepting of gender nonconformity (compared with other cisgender controls). Could this reflect some differential parental socialization taking place with regard to acceptability of deviations from gender norms? There is also no explanation regarding the value of considering both parents’ actual gender stereotyping vs. adolescents’ perceptions of the extent to which their parents would gender stereotype. In short, the Introduction should be developed further to convey the theoretical and/or practical importance of the kinds of questions being asked here. To the extent that directional predictions (even if competing) can be offered, I encourage that as well.

Thank you for this suggestion. We have scaffolded the introduction with more relevant discussion of past work that speaks to the inclusion of these measures and research questions. In particular, we have added more background on parental gender socialization as it informs hypotheses about whether parents of transgender or cisgender adolescents might show more or less explicit stereotype endorsement. In addition, we have added text that clarifies why we measured adolescents’ perceptions of the extent to which their parents would endorse gender stereotypes. While we did not have a priori directional predictions, we provide information about why one might speculate different patterns of results given prior literature.

2. I found that there were several statements of fact that were not backed up by a citation. Examples include claims that the numbers of trans adolescents are increasing and that past work generally finds that parents influence their children’s thinking about gender (regarding this latter one, I wonder whether the authors meant children and adolescents because the latter seems more relevant to concentrate on here). I suggest the authors make sure to back up statements of fact with appropriate citations.

Thank you for this suggestion; we have made sure that all broad statements in the article are supported by prior literature.

3. What is the benefit of retaining participants who skipped all questions? Seems that if they did not provide data on the dependent variables, they should be dropped because they are leading to a skewed sense of the characteristics of the groups of participants who contributed the key data.

We apologize for the error in reporting. Participants who completed no items were not included in analyses. (However, adolescents who completed the self-report items but none of the items predicting the parent’s stereotyping were included in the applicable analyses.)

4. The samples do not seem to be matched on demographics. The cisgender sample appears to be more affluent and less white/more multiracial (based on the parent data). Are the authors concerned that this might be an important confound in their analyses? There is some research suggesting racial differences in gender development exist (e.g., work by May Ling Halim). Also, the cisgender controls were recruited from a participant database at the authors’ institution. In the present case, I assume this means that participants were from a relatively urban center in the US Pacific Northwest, which probably has a particular social climate. I wonder whether the gender diverse sample is from a more varied set of backgrounds given they were recruited from across the US and Canada, and whether this is also a relevant confound to consider in weighing the comparability of the participant groups.

Thank you for bringing up this point. The samples are not perfectly matched on demographics. We have included chi-squared tests and two-sample t-tests and results in the participant demographics table to illustrate differences between the two groups. Nonetheless, given the general lack of difference on the dependent measures between the groups, we think it is unlikely that demographic differences between the two groups played a large role. However, we have added text to the limitations section highlighting that the two groups are not matched on demographics and that the biases, particularly in the cisgender group, might explain why we observed so little stereotyping. 

5. Can the authors please say more about the recruitment method for online participants as well as for the other samples? I know this is part of a larger project and the authors have maybe shared some of these details elsewhere (especially on the longitudinal sample), but it’s not clear for those who are maybe only reading this paper from this team. Also, I wonder how this sample relates to prior ones that this group reported on with respect to gender stereotyping. There were three relevant studies from this team that were reviewed in the Introduction. Were the participants in this study the same as any of those prior ones? Or is this a completely different cohort?

Thank you for this suggestion. We have included text that clarifies (a) how participants from the Trans Youth Project cohort were recruited (n=79), (b) how many of the gender diverse children in this paper have been profiled in prior studies (n=8), and (c) how gender diverse participants who were not part of the Trans Youth Project cohort were recruited (n = 64).

6. The authors do not note the parent gender. There is literature suggesting mothers and fathers hold different attitudes about gender roles/stereotypes. I suggest reviewing that literature and analyzing by parent gender. Perhaps see work by Joyce Endendijk and colleagues on this topic.

Thank you for this suggestion. We have included text in the Results section which indicates how many parents of each gender there were in each group, as well as an exploratory analysis examining whether there were differences in stereotyping between parents who identified as men and parents who identified as women. We have also included text citing literature about how mothers and fathers hold different attitudes about gender roles and stereotypes, including work by Endendijk.

7. The authors do not note the adolescent participants’ gender breakdown. Are there adolescent gender differences among cis samples in gender stereotyping? Any reason to suspect there might be differences between trans boys vs. trans girls vs. nonbinary, and so on? Even a preliminary analysis of this question would be important/interesting. In any case, it is presently unclear how comparable the samples are with respect to the gender composition with regard to cis/trans feminine/masculine individuals.

Thank you for this suggestion. We have included information about how many adolescents of each gender there were in each group (Table 2) and the means and standard errors on the stereotyping measure for adolescents of different genders (Table 5). Further, we have included an ANOVA that tests for differences in stereotyping by gender. 

8. Can the authors please provide reliability analysis data on the OAT-AM for the present sample? Are all the items on this scale contributing to reliability? For example, there is some recent work suggesting that the stereotype that boys are superior at math is not always endorsed in adolescent samples (Morrissey et al 2019 in J Adolescence).

Thank you for this suggestion. In the results section, we have included Crohnbach’s alpha values in the results for each of the three iterations of the trait subscale of the OAT-AM that we administered. We do not include the full analysis showing how much item contributed to reliability; there were no items that brought down reliability of the scale more than .03. We do note, however, that reliability on the parent self-report measure is lower than the others, and discuss potential reasons for the low variability.

9. The authors state that the OAT-AM used in the current study was adapted lightly. Please explain.

We have clarified that the trait subscale of the OAT-AM was adapted by removing the “Neither men nor women” option and adding a “skip” option for each item.

10. It would be helpful to explain how the OAT-AM was scored in the Method section. A statistical analysis subsection would also be helpful to evaluate/understand the analytic approach. As is, the research questions and analyses are all somewhat vague, which makes it difficult to discern whether the optimal approach is being employed.

Thank you for this suggestion. We have clarified in the Methods section how the OAT-AM is scored. Additionally, we have revised the results section around 3 questions, which are hopefully more clear than they were in the first submission. 

● The first research question is whether participants showed gender stereotyping at all, without considering group differences. To answer this question, we include a one-sample t-test to test whether the mean value across all participants in the gender-stereotyping measure is different from 3 (the response which corresponds to ‘no gender stereotyping’). 

● The second research question is whether there are group differences (both cisgender vs. gender diverse, as well as adolescents vs. parents) in gender stereotyping endorsement. To answer this question, we fit a mixed-effects linear regression predicting an individual’s mean stereotyping score from (1) whether they are an adolescent or a parent and (2) whether they are from a family with a cisgender teen participant or a family with a gender diverse teen participant.

● The third research question is whether adolescents’ gender stereotype endorsement, and their assumptions about their caregivers’ gender stereotype endorsement are predictive of parents’ gender stereotype endorsement. To that effect, we conduct linear regressions that examine the strength of the relationship between adolescents’ responses (of both types) and their parents’ responses. 

11. Tables 3: Please provide a more descriptive title.

We have provided more descriptive titles.

12. A main finding is that participants, regardless of group or age, were unlikely to endorse prescriptive gender stereotypes. One wonders what might have happened had the authors measured descriptive stereotypes. I also wonder whether we are now in an era where people hold (or at least report) explicit views that run contrary to traditional gender stereotypes. Perhaps an implicit measure would yield some different results. Given this team’s expertise in this area, I would be interested to see some discussion of these possibilities folded into this paper.

Thank you for this suggestion. We have included text in the discussion section that suggests these areas as future topics of interest. We agree that they may be more productive than this focus on prescriptive work moving forward (though we didn’t know that before we found these results.)

13. Page 8, line 255: The authors are making the point that one study found trans children ages 6-8 years gender stereotyped less. But in the Introduction they noted that trans children that study were similar to their cis siblings. So, it seems a little dubious to me to claim in the Discussion that the study of 6-8 year-olds is finding something that suggests a trans vs. cis difference.

Great point. We have amended the language to indicate that trans youth and their siblings differed from unrelated cisgender youth in their stereotyping. 

14. The figure quality appeared “fuzzy” on my end. Consider revising to higher resolution.

Thank you for this suggestion. We have made sure the figure appears in higher resolution.

---

## [Editor Report · Decision Letter 1]

28 Feb 2022

PONE-D-21-20601R1

Endorsement of gender stereotypes in gender diverse and cisgender adolescents and their parents

PLOS ONE

Dear Dr. deMayo,

Thank you for submitting your manuscript to PLOS ONE. I have now had the opportunity to read your revised manuscript and I believe that you have done a good job of addressing the majority of issues raised in the first round of reviews. As such, I have decided not to send it back out for review, as I feel that it has merit but does not fully meet PLOS ONE’s publication criteria as it currently stands. Instead, I would like to provide you with the opportunity to engage in one final round of revisions and I will hope to make a decision about the manuscript soon after receiving this revision. Please be sure to submit and correspondence directly through the PLOS ONE system to ensure timely responses; my apologies for delays in receiving a decision about this revision as I only received it through the system recently.My main concern remains whether this provides sufficiently original research to warrant publication in this journal. I believe that it does, owing largely to the need for additional literature examining the social cognition of transgender youth. This also includes stereotyping by both youth and their parents, as well as youth’s expectations around their parents’ results. This is impressive. One additional question that is not addressed, and that I believe could help to bolster the novelty of this research, would be to also examine male and female stereotypes separately. That is, are there differences for feminine stereotypes or masculine stereotypes? Would you expect differences between these groups on either? This of course would need to be noted as a post-hoc analysis but might add to the substance of the article and possible the findings (e.g., in the case where no differences emerge). Similarly, on page 14 you suggest that more stereotyping occurred for personality traits versus academic domains. Additional analyses could examine this statistically as a post-hoc analysis (although you would need to note that should be interpreted with caution due to the post-hoc nature of this type of analysis). One or both of these might be most appropriately added to the supplement as opposed to the main text and I will leave that to you to decide. In short, I felt that some interesting questions were left unaddressed and addressing them would help to contribute the concerns previously raised by Reviewer 2 regarding the contribution of this article.  I was a bit confused by some of the information in Table 1. For one, I believe the chi-squared tests are comparing two groups (and show difference between (not among) groups?). For gender, I assume this is between woman and men with other/not reported being excluded from the analysis? For yearly income, it was not clear to me what was being compared. Please clarify.In the exclusions section, it would seem that after these exclusions were made, the total number of participants listed in the Participants section remained. However, given the placement of this, it was not clear (until the end of the section) whether these were exclusions beyond the N=145. Please start by listing the total number of participants run; that is: We started with a total of X participants; however, “during data collection, we noticed…” to clarify.Please provide a table of correlations that corresponds to the regression analyses.  

Additional considerations:  

Stylistically, I would recommend removing the ‘road map’ on page 4 and instead relying on your headers to guide the reader.On page 7 you seem to suggest that an advantage of this measure is that it has “often produced clear indications of gender stereotyping in this age group”. I would rephrase as it is not clear from this wording why this is an advantage. I see this instead as a previous finding that is relevant to your research questions and predictions.I was not clear about the dfs provided for the F-statistic in the first paragraph of page 18. Also, you note that this compares boys, girls, and non-binary/other adolescents, but in the discussion, you note findings regarding boys and girls. If this is an omnibus F with three groups please provide post-hoc direct comparisons.To decrease possible confusion, I would recommend referring to tables in the supplement as Table S1, etc.Although you list all ten traits in the supplement, I think that it would also be helpful for readers to see these listed in the manuscript on page 13.On page 14, you note that more people skipped specific items; you might include those ns in Table S7.  Overall, I find that you have presented some interesting findings. I think that this has the potential to make an important contribution to our understanding of gender stereotyping in adolescence. I hope that you will decide to revise and resubmit this paper and will look forward to receiving your re-submission. I will hope to make a quick decision after receiving these revisions.

We look forward to receiving your revised manuscript.

Warmly,

Jennifer Steele

Academic Editor

PLOS ONE
---

## [Author Response · Author response to Decision Letter 1]

25 Apr 2022

Dear Dr. Steele,

Thank you so much for allowing us to revise and resubmit our manuscript, “Endorsement of gender stereotypes in gender diverse and cisgender adolescents and their parents”. 

The most notable changes to the manuscript include:

We have added analyses probing whether (a) masculine and feminine stereotypes are endorsed to differing extents and (b) whether stereotypes in different domains - academic vs. personality - are endorsed to differing extents. Specifically, as we describe below, we include tables in the manuscript that show means and standard deviations of stereotyping scores broken down on both of these dimensions, and refer readers to the supplementary materials for more detailed statistical inference probing effects related to stereotype gender and domain.

We have added more information about the measure - most notably, a table with all of the items and their means and standard errors - into the manuscript itself.

Below, you will find a point-by-point response to comments from your most recent review of this manuscript.

Thank you again so much for your time in reviewing this work and allowing us to resubmit. We look forward to hearing your decision.

Sincerely,

Benjamin deMayo

Shira Kahn-Samuelson

Kristina Olson

My main concern remains whether this provides sufficiently original research to warrant publication in this journal. I believe that it does, owing largely to the need for additional literature examining the social cognition of transgender youth. This also includes stereotyping by both youth and their parents, as well as youth’s expectations around their parents’ results. This is impressive. One additional question that is not addressed, and that I believe could help to bolster the novelty of this research, would be to also examine male and female stereotypes separately. That is, are there differences for feminine stereotypes or masculine stereotypes? Would you expect differences between these groups on either? This of course would need to be noted as a post-hoc analysis but might add to the substance of the article and possible the findings (e.g., in the case where no differences emerge). Similarly, on page 14 you suggest that more stereotyping occurred for personality traits versus academic domains. Additional analyses could examine this statistically as a post-hoc analysis (although you would need to note that should be interpreted with caution due to the post-hoc nature of this type of analysis). One or both of these might be most appropriately added to the supplement as opposed to the main text and I will leave that to you to decide. In short, I felt that some interesting questions were left unaddressed and addressing them would help to contribute the concerns previously raised by Reviewer 2 regarding the contribution of this article. 

Thank you for these suggestions. Regarding originality, a few aspects of this work that we believe are an original contribution are: this is the first work on gender stereotyping in transgender adolescents (that we know of!), this is a contemporary test of a “classic” and common measure – that perhaps suggests the measure is getting to be less useful (thereby suggesting researchers might opt for others in the future), and this work is original in its inclusion of teens and their parents to look for within family associations.

In addition, we agree that the suggested analyses further increase the novelty of the work. In the “Gender stereotype endorsement” section of the Results (page 17), we include tables showing the means and standard errors for stereotyping scores, broken down both by the gender of the stereotypes in the items (masculine vs. feminine) and the domain of the stereotype (academic- vs. personality- related). We note that mean scores are higher for personality-related and feminine stereotypes, and point readers to a more detailed analysis examining these effects in the supplement (S5 and S6). 

As you noted, we did not preregister a specific prediction for differential endorsement of masculine vs. feminine stereotypes, though some literature suggests that people are generally more comfortable with girls showing masculine stereotypes than the reverse (e.g., Martin, 1990; Coyle, Fulcher, & Trubutschek, 2016), which is consistent with our finding that people endorsed feminine stereotypes as being more for girls (only) than masculine stereotypes were for boys (only).

I was a bit confused by some of the information in Table 1. For one, I believe the chi-squared tests are comparing two groups (and show difference between (not among) groups?). For gender, I assume this is between woman and men with other/not reported being excluded from the analysis? For yearly income, it was not clear to me what was being compared. Please clarify.

Thank you for this suggestion; we have clarified what is being compared in each of the tests in Table 1. Specifically, we added notes that specify what is being compared in each of the chi-squared tests in both Table 1 and Table 2. For the chi-square tests comparing race, we are comparing white or non-white groups (because the small cells of specific groups violate assumptions of the chi-square analyses). For the gender comparison, we bin men together with “other/not reported” since participant N’s for men and “other/not reported” are so low. For the yearly income comparison, we converted participants’ reported income categories to a 1-5 scale (1 being the lowest, 5 being the highest) and ran a t-test comparing mean income scores between the gender diverse and cisgender group parents.

In the exclusions section, it would seem that after these exclusions were made, the total number of participants listed in the Participants section remained. However, given the placement of this, it was not clear (until the end of the section) whether these were exclusions beyond the N=145. Please start by listing the total number of participants run; that is: We started with a total of X participants; however, “during data collection, we noticed…” to clarify.

Thank you for this suggestion. We have restructured the Participants section slightly so that we include exclusion-related information for each subgroup of participants in our study (gender diverse adolescents and their parents, and cisgender adolescents and their parents). For each subgroup, we describe how many responses we received, how many we excluded for specific reasons, and how many participants were left in the final sample.

Please provide a table of correlations that corresponds to the regression analyses.

We were not exactly sure what table of correlations you are seeking, but we do now report a table of the Pearson’s r correlations between the three main stereotype measures (adolescent self-report, parent self-report, and adolescent prediction about the caregiver) in Supplementary Material 7. Apologies if this is not what you were seeking. 

If you were seeking correlations between our categorical predictors (gender diverse vs. cisgender, parent vs. adolescent), the analogous information is captured in the regression summarized in Table 6, which examines group differences in self-reported stereotyping across in both adolescents and parents in the gender diverse and cisgender groups. 

Similarly, if you were seeking information about how the relationship between parents’ and adolescents’ stereotyping might be different in the gender diverse vs. cisgender dyads, this information is captured in Tables S5 and S6 in the Supplementary Materials. 

If you meant to suggest a different set of correlations, please let us know and we would be happy to add it.

Stylistically, I would recommend removing the ‘road map’ on page 4 and instead relying on your headers to guide the reader.

Thank you for this suggestion; we have removed the road map text and instead use conceptual headers throughout.

On page 7 you seem to suggest that an advantage of this measure is that it has “often produced clear indications of gender stereotyping in this age group”. I would rephrase as it is not clear from this wording why this is an advantage. I see this instead as a previous finding that is relevant to your research questions and predictions.

Thank you for this point. We have removed the text in this paragraph that lists this as an advantage of the measure. Instead, we highlight the results observed in past studies using the trait subscale of the OAT-AM. 

I was not clear about the dfs provided for the F-statistic in the first paragraph of page 18. Also, you note that this compares boys, girls, and non-binary/other adolescents, but in the discussion, you note findings regarding boys and girls. If this is an omnibus F with three groups please provide post-hoc direct comparisons.

Thank you for raising this point. The F statistic refers to a one-way ANOVA comparing mean stereotyping scores among boys, girls, and nonbinary/other adolescents; thus, there were three groups, and 2 degrees of freedom. In addition, as you suggested, we have added post-hoc pairwise tests as well, which match the finding pointed out in the discussion.

To decrease possible confusion, I would recommend referring to tables in the supplement as Table S1, etc.

Thank you, we have adjusted how we refer to tables in the supplement.

Although you list all ten traits in the supplement, I think that it would also be helpful for readers to see these listed in the manuscript on page 13.

Thank you. We now include a table with all the items (including their means, standard errors, and the number of participants who skipped each) in the “Gender stereotype endorsement” subsection of the Results section (Table 3). 

On page 14, you note that more people skipped specific items; you might include those ns in Table S7.

Thank you for this suggestion - we have added participant N’s for those who skipped each item in the table mentioned above (Table 3 in the manuscript).

---

## [Editor Report · Decision Letter 2]

31 May 2022

Endorsement of gender stereotypes in gender diverse and cisgender adolescents and their parents

PONE-D-21-20601R2

Dear Dr. deMayo,

I believe that you have appropriately addressed each of my outstanding comments and I am pleased to inform you that your paper is being accepted for publication. I think that this paper will make a nice contribution to the field and want to commend you on this work. I can confirm that your manuscript has been judged scientifically suitable for publication and will be formally accepted for publication once it meets all outstanding technical requirements.

I want to thank you again for considering PLOS ONE as an outlet for this research and want to wish you all the best in your future research endeavors.

Warmly,

Jenn Steele

Academic Editor

PLOS ONE

---

## [Editor Report · Acceptance letter]

6 Jun 2022

PONE-D-21-20601R2 

Endorsement of gender stereotypes in gender diverse and cisgender adolescents and their parents 

Dear Dr. deMayo:

I'm pleased to inform you that your manuscript has been deemed suitable for publication in PLOS ONE. Congratulations! Your manuscript is now with our production department. 

Kind regards, 

on behalf of

Dr. Jennifer Steele 

Academic Editor

PLOS ONE